# CTS-Net: A Segmentation Network for Glaucoma Optical Coherence Tomography Retinal Layer Images

**DOI:** 10.3390/bioengineering10020230

**Published:** 2023-02-08

**Authors:** Songfeng Xue, Haoran Wang, Xinyu Guo, Mingyang Sun, Kaiwen Song, Yanbin Shao, Hongwei Zhang, Tianyu Zhang

**Affiliations:** Key Laboratory of Geophysical Exploration Equipment, Ministry of Education, College of Instrumentation and Electrical Engineering, Jilin University, Changchun 130000, China

**Keywords:** glaucoma, deep learning, loss function, retinal layer segmentation, optical coherence tomography

## Abstract

Optical Coherence Tomography (OCT) technology is essential to obtain glaucoma diagnostic data non-invasively and rapidly. Early diagnosis of glaucoma can be achieved by analyzing the thickness and shape of retinal layers. Accurate retinal layer segmentation assists ophthalmologists in improving the efficiency of disease diagnosis. Deep learning technology is one of the most effective methods for processing OCT retinal layer images, which can segment different retinal layers and effectively obtain the topological structure of the boundary. This paper proposes a neural network method for retinal layer segmentation based on the CSWin Transformer (CTS-Net), which can achieve pixel-level segmentation and obtain smooth boundaries. A Dice loss function based on boundary areas (BADice Loss) is proposed to make CTS-Net learn more features of edge regions and improve the accuracy of boundary segmentation. We applied the model to the publicly available dataset of glaucoma retina, and the test results showed that mean absolute distance (MAD), root mean square error (RMSE), and dice-similarity coefficient (DSC) metrics were 1.79 pixels, 2.15 pixels, and 92.79%, respectively, which are better than those of the compared model. In the cross-validation experiment, the ranges of MAD, RMSE, and DSC are 0.05 pixels, 0.03 pixels, and 0.33%, respectively, with a slight difference, which further verifies the generalization ability of CTS-Net.

## 1. Introduction

Glaucoma is an eye disease that can lead to blindness. Early diagnosis and treatment with ophthalmologists can prevent further deterioration [1]. It is estimated that the number of glaucoma patients worldwide will increase from 76.5 million in 2020 to 111.8 million in 2040 [2]. Glaucoma is characterized by thinning of the retinal nerve fiber layer (RNFL) and optic disc depression. The main factors that form glaucoma are age, elevated intraocular pressure (IOP), and genetic background [3]. The main parameter for early diagnosis of glaucoma is the thickness of the RNFL layer: the smaller the thickness, the more severe the symptoms [4]. Thanks to the non-invasive, fast scanning speed, high resolution, and 3D imaging advantages of OCT technology [5], it soon became a necessary technical means for diagnosing ophthalmic diseases [6]. Ophthalmologists analyze OCT images to determine eye or body health conditions, such as glaucoma, multiple sclerosis [7], and Alzheimer’s [8]. Different tissue layers of the retina have strict topological edge order. OCT retinal layer image segmentation results are significant for thickness [4,9] and surface shape analysis [10]. However, images collected by OCT technology are usually rough due to noise, and the layer boundary is unclear. Ophthalmologists must carefully analyze retinal OCT images to identify the retinal layer and its edge, which usually takes plenty of time. From the above, there is an urgent need for OCT automatic segmentation technology.

Many automatic retinal layer segmentation methods have been proposed to help doctors analyze OCT images. Their main goal is to obtain the correct and smooth retinal layer surface. Graph-based methods [11,12,13], which only use hand-designed features, are vulnerable to the noise and distortion of OCT images. In recent years, deep learning has developed rapidly and is widely used in the medical field [14]. The U-shaped [15] neural network framework based on convolutional neural network (CNN) is the most widely used in ophthalmic retinal layer segmentation [16], e.g., Ref. [17] uses a fully convolutional neural network to predict each OCT image pixel and then extract the edge, Ref. [18] divides the pixel into ten categories and applies the shortest path to obtain the edge of each layer, and Ref. [19] uses ResUnet to directly predict the retinal layer category of each OCT image’s pixel and each layer’s edge position. Although the CNN-based model achieves good performance, it cannot learn the interaction of semantic information between global and long range due to the limitation of the convolution operation. On the contrary, the model constructed based on the Transformer has the ability of global modeling, and its performance surpasses the model based on CNN [20]. Among them, the modified model based on Transformers such as Swin Transformer [21] and CSWin Transformer [22] have outstanding performance in terms of accuracy. The model based on the Transformer achieves satisfactory performance, but there are fewer applications in OCT images in retinal layer segmentation.

In order to apply the excellent performance of the Transformer to the segmentation of retinal layers, our network adopts a design combining convolution and the Transformer. Since the Transformer’s self-attention calculation consumes a lot of computing power [23] and cannot directly process image data, it is necessary to use the convolutional layer to convert the image into sequence data containing multiple tokens. The backbone layer used by our network is the CSWin Transformer [22], which enables powerful modeling capabilities while constraining computational cost; this is meaningful for improving the convergence speed of model training and small dataset training. Our network model follows the classic U-shaped structure design to enable the model to learn more features at multiple scales. We apply the proposed method to the retinal layer segmentation data, which is an essential reference for glaucoma, and compare it with the state-of-the-art method to verify the effectiveness of our method. Inspired by [24], we propose a Dice loss function based on the edge area and use it for neural network training. The results show that this is effective for improving the characteristics of the edge area learned by the neural network and the segmentation accuracy.

Our contributions can be summarized as follows:We design a CSWin-Transformer-based OCT image segmentation network for glaucoma retinal layers. After carefully analyzing the cross-attention mechanism of the CSWin Transformer, it is found that its self-attention in the horizontal and vertical directions matches the features of the retinal layer. Therefore, we developed the neural network and applied it to the segmentation task of glaucomatous retinal layers, which provides a new reference direction for using an attention mechanism for retinal layer segmentation.We present a Dice loss function based on edge regions. In retinal layer segmentation tasks, features are often condensed in edge regions. Based on the Dice loss function, we developed a loss function that only calculates the overlapping loss of the edge region, which can guide the depth learning model to learn more edge features to improve the accuracy of edge segmentation.

## 2. Related Works

### 2.1. Retinal Layer Segmentation

In the images of the retina layer collected by OCT technology [5], the change in gray intensity depicts the tissue characteristics of different retina layers, such as the nerve fiber layer (NFL), ganglion cell layer (GCL), inner plexiform layer (IPL), inner nuclear layer (INL), outer plexiform layer (OPL), outer nuclear layer (ONL), inner segment outer segment junction (ISOSJ), outer segment layer (OSL), outer segment photoreceptors (OPR), subretinal virtual space (SRVS–zero thickness in normals), and retinal pigment epithelium (RPE) [25]. The thickness change of a single layer may be a precursor of retinal disease, which is of great significance for early clinical diagnosis. For example, glaucoma is an optic nerve disease that causes irreversible vision loss, resulting in loss of nerve fibers (i.e., thinning of the RNFL) and depression of the optic nerve head [2]. Deep learning is a complex algorithm in machine learning, which has strong learning ability, robustness, and can obtain the desired results from data with complex noise. Although deep learning is computationally intensive and requires many data, it is relatively affordable for modern technology. It has achieved excellent results in language translation and image recognition [26] and has also been widely used in pathological recognition and tumor segmentation in the medical field [4,14]. CNN-based deep learning technology has been widely used in analyzing fundus OCT images [27]. However, there are few retinal segmentation networks based on CSWin Transformer.

### 2.2. Transformer

Proposed initially by [28], the Transformer was applied to natural language processing and made significant progress. Inspired by the achievements of the Transformer, researchers applied it to the field of computer vision [23] and designed the Vision Transformer (ViT). Previously, the basic framework of the deep learning model used for computer vision tasks was CNN; however, now, the model using the Transformer as the basic structure has achieved better results [20]. Transformer-based models have been widely used in visual tasks such as image classification, object detection, semantic segmentation, and video understanding. Their excellent results are due to the attention mechanism. The deep learning model draws on the human attention mechanism, which quickly screens high-value information from a large amount of information using limited attention resources. Various attention mechanisms have been proposed, such as sparse attention, linearized attention, and multi-head attention [29]. Among them, multi-head self-attention is the most widely used, and the representative model for this mechanism is Swin Transformer. Swin Transformer proposes a moving window attention mechanism to solve the computationally intensive problem of self-attention on the entire image. It divides the image into windows of different sizes and performs self-attention calculations separately. In order to increase the attention span but reduce the amount of calculation, Ref. [22] proposes a cross-attention mechanism, which divides the image into non-overlapping stripes in both horizontal and vertical directions and performs self-attention calculations in each stripe. Compared with the moving window attention mechanism, the cross-attention mechanism can achieve a more comprehensive attention range and realize parallel computing of attention. Inspired by the operation method of the cross-attention mechanism and combined with the hierarchical feature analysis of the OCT retina, we found that this segmentation task is suitable for applying the cross-attention mechanism, which can perform self-attention calculations on the retinal layers in the horizontal and vertical directions. This is believed to help improve the accuracy of image segmentation. Therefore, we desire to utilize the CSWin Transformer to design a neural network model.

### 2.3. Loss Function

Various deep learning models have been proposed and applied to different fields, among which training a well-performing model is inseparable from the loss function used. A loss function measures the degree of difference between the prediction and the ground truth, and the loss result guides the model learning through the backpropagation mechanism of the neural network. In medical image segmentation, the generally used loss functions of neural network models are the cross-entropy loss function and Dice loss function [24]. Cross-entropy comes from information theory and is used in machine learning to evaluate the class probability distribution of the model output and the accurate distribution. The dice coefficient was initially used to measure the coincidence of two images. Milletari et al. [24] improved the dice coefficient into a loss function and applied it to the MRI image segmentation task of the prostate. We assign priority to the edge regions in the retinal layer segmentation task. To enable the model to learn more edge area features of retinal OCT images, we improved the Dice loss function, called BoundaryAreaDiceLoss, which only selects an area of a certain width at the edge of each layer to calculate the loss. Then, using BoundaryAreaDiceLoss in combination with other loss functions (such as Dice Loss), the model will obtain more gradients from the edge area of each category during training—that is, assign priority to the features of the edge area.

## 3. Method

### 3.1. Network Architecture

As shown in Figure 1, our neural network model includes an encoder, decoder, and skip connections. The backbone network used by our model is the CSWin Transformer Block (CTB). For the encoder, the original image is processed by Pre-processing (PreP) to obtain multiple square images that are of the same size and non-overlapping; these images are input to the CC layer as a batch. Following the method of [22], the sub-layer ConvEmbed layer in the CC layer operates overlapping convolution (kernel size is 7 × 7, the stride is 4) to convert the picture into a sequence. The number of channels can be arbitrarily amplified to C. Then, via the Patch Merging (PM) layer and the continuous CTB layer, we can obtain multi-scale feature representation. The PM layer is responsible for downsampling to expand the receptive field, and the CTB is used for feature representation learning. The decoder upsamples deep-scale features through three network layers (PEC, CC, and PE) and then fuses shallow elements through skip connections. Multiple feature maps are generated by the PEC layer and then input to the CC layer for further feature analysis, and double upsampling is performed through the PE layer again. The last layer of PEC outputs a pixel-level prediction map for each retinal layer. After Post-processing (PostP), the image with the input size of the original image is obtained. PreP and PostP will be explained in Section 4.2.

### 3.2. CSwin Transformer Block

The CSWin transformer block (CTB) consists of 2 layers of LayerNorm (LN), cross-shaped window self-attention, multi-layer perceptron (MLP), and skip connections. As shown in Figure 2. CTB is built based on the cross-shaped windows attention mechanism, its input and output dimensions are consistent, and the operation process is expressed by Equation (Equation 1).
(1)$$\begin{array}{c} {\hat{X}}^{l}=\mathrm{CSWin}-\mathrm{Attention}(\mathrm{LN}(X^{l-1}))+X^{l-1}\;\\ X^{l+1}=\mathrm{MLP}\left(\mathrm{LN}\left({\hat{X}}^{l}\right)\right)+{\hat{X}}^{l} \end{array}$$
where ${\hat{X}}^{l}$ is obtained by splicing the output of Cross-Shaped Window Self-Attention and the input $X^{l-1}\in R^{(H\times W)\times C}$. $X^{l+1}$ is obtained by concatenating the output of the MLP layer with ${\hat{X}}^{l}$.

The cross-window self-attention mechanism achieves efficient global attention by simultaneously performing self-attention computations in both horizontal and vertical directions. When performing attention calculations, the image is divided into multiple stripes of the same size in the two directions, and the width of the stripes can be customized. The stripe width is an essential parameter of this attention module: the larger it is, the more connection between categories can be established in a more extensive range, but the amount of calculation will increase slightly. The horizontal attention calculation is expressed by Equation (Equation 2), and the final attention calculation result is defined by Equation (Equation 3).
(2)$$\begin{array}{c} X=[X^{1},X^{2},\ldots ,X^{M}]\;\\ Y_{k}^{i}=\mathrm{Attention}\left(X^{i}W_{k}^{Q},X^{i}W_{k}^{K},X^{i}W_{k}^{V}\right)\;\\ H-\mathrm{Attentio}n_{k}\left(X\right)=\left[Y_{k}^{1},Y_{k}^{2},\ldots ,Y_{k}^{M}\right] \end{array}$$
where $X^{i}\in R^{(sw\times W)\times C}$, M=H/sw, and i=1,…,M. $W_{k}^{Q}\in R^{C\times d_{k}}$, $W_{k}^{K}\in R^{C\times d_{k}}$, and $W_{k}^{V}\in R^{C\times d_{k}}$ represent the query, key, and value matrix of the $k^{th}$ head, respectively, and each head has dimension $d_{k}$.
(3)$$\begin{array}{c} \mathrm{CSwin}-\mathrm{Attention}(X)=\mathrm{Concat}(\mathrm{hea}d_{1},\ldots ,\mathrm{hea}d_{k})W^{O}\;\\ \mathrm{hea}d_{k}=\left\{\begin{array}{cc} H-\mathrm{Attentio}n_{k}\left(X\right) & k=1,\ldots ,K/2\\ V-\mathrm{Attentio}n_{k}\left(X\right) & k=K/2+1,...,K \end{array}\right. \end{array}$$
where $W^{O}\in R^{C\times C}$ is the projection matrix that maps self-attention results to the target output dimension (*C*).

### 3.3. Encoder

The encoder contains three network layers: CC, PM, and CTB. A batch of original images is obtained through Pre-processing and input to the CC layer. ConvEmbed comes from [22], which can convert images into sequence data for continuous CTB learning feature representation. After the PM [21] layer divides the input patches into 4 parts and connects them, the number of channels increases to 4×. Then, PM uses a linear layer to change the number of channels to 2× of the original input patches. Then, the representation learning of this scale is performed through 4 consecutive CTBs. In the encoder, the calculation process of PM and CTBx4 will be repeated twice. Thanks to CTB’s cross-attention mechanism, the encoder can perform self-attention calculations in an extensive range in the height and width directions of the image, which is beneficial for learning the hierarchical features of the OCT fundus images.

### 3.4. Decoder

Each decoder block consists of PEC, CC, PE, and CTB, where the structure of CC is the same as mentioned in the Encoder section. The PEC layer contains PE and convolutional layers (Conv). The PE [21] layer uses the linear layer to 2× upsample the input patches. Conv restores the input patches to the feature map of the current scale. The first PEC layer outputs a feature map of *N* categories and size $\frac{H}{8}\times \frac{W}{8}$ from the deep features. The PEC, CC, and PE are equivalent to feature extraction and feature learning at the current scale and then perform feature fusion on the shallow features passed by skip connections through two consecutive CTBs. After repeating two decoder blocks, the feature map size of the final PEC layer output is H×W.

### 3.5. Loss Function

Inspired by the Dice loss function proposed in [24] and combining the OCT fundus image segmentation task with high requirements for edge segmentation, we designed a Dice loss function based on the edge area, named BoundaryAreaDiceLoss ($L_{BADice}$). This loss function can guide the model to learn more about the edge area features of the retinal layer to improve the accuracy of edge segmentation. As shown in Figure 3a, cyan is the prediction $P_{i}$ of the *i*-th category and green is the ground truth $G_{i}$ for which we take the area whose edge width is *d*, as shown in Figure 3b, to calculate the Dice loss. $L_{BADice}$ is represented by Equation (Equation 4).
(4)$$L_{BADice}=\frac{1}{N}\sum_{i=0}^{N-1}\left(1-\frac{2|P_{i}^{ba}\cap G_{i}^{ba}|+\epsilon }{|P_{i}^{ba}\left|+\right|G_{i}^{ba}|+\epsilon }\right)$$
where *N* is the number of categories. $P_{i}^{ba}$ and $G_{i}^{ba}$ are the predicted and ground-truth results for the *i*-th class of edge regions, respectively. In order to prevent the situation where both the numerator and the denominator are 0, a constant ϵ is added.

To ensure precise prediction of the middle area of each category, we use the Dice loss function ($L_{Dice}$) and BoundaryAreaDiceLoss ($L_{BADice}$) jointly; then, the loss function of the final network training is
(5)$$L=w_{1}*L_{Dice}+w_{2}*L_{BADice}$$
where $w_{1}$, $w_{2}$ are the weighted hyperparameters.

## 4. Experiments

### 4.1. Dataset

We applied the proposed method to a segmentation competition dataset GOALS [30], provided by the Zhongshan Ophthalmology Center of Sun Yat-sen University in Guangzhou, China. We utilized 100 sample data obtained by peripapillary circular scanning, half of which were glaucoma patients. The label image size is 800 × 1000, with segmentation marks of the retinal nerve fiber layer (RNFL), ganglion cell plexiform layer (GCIPL), and choroid layer (CL).

### 4.2. Image Processing

#### 4.2.1. Pre-Processing

Due to the high resolution of sample images and the limitation of GPU memory, the batch size must be reduced in the training stage. If the batch size is too small, the convergence of the model will be unstable [31], and if it is too large, the convergence will be slow [32]. To solve this problem, we intercept the range of pixel index 63–613 in the length direction of the original image, which can include all retinal layer areas to obtain a new length size of 550 and then downsize it to 256. Meanwhile, the width direction is directly reduced to 1024. The newly generated size is 256 × 1024. Then, divide the image into four non-overlapping parts, each with a size of 256 × 256. The processing flow is shown in Figure 4.

The processing of labels requires further improvement based on the process shown in Figure 4. The retinal layer area we want to segment has three layers, as mentioned in Figure 5b, but there are five retinal layer edges. To reduce the impact of background pixels occupying the majority and resulting in highly unbalanced categories, we divide the pixels into six categories according to the number of boundaries.

#### 4.2.2. Post-Processing

The output of the proposed neural network in the last PEC layer is the pixel-level segmentation prediction map of each sub-eye layer. However, the edges between layers need further processing to obtain. We first use the Canny algorithm [33] to extract edge pixels, which will result in jagged edges because pixels cannot be further divided. These edges are then fitted by the Savitzky–Golay [34] algorithm to obtain smooth retinal layer edges. The Savitzky–Golay filtering algorithm is a window-based data weighting filter that performs polynomial fitting according to the least squares method. It can improve the accuracy of the data without changing the signal trend and width, which can be expressed by Equation (Equation 6). We set the window size to 33, and the result is shown in Figure 6.
(6)$$x_{k,smooth}={\bar{x}}_{k}=\frac{1}{H}\sum_{i=-w}^{+w}x_{k+i}h_{i}$$
where $x_{k,smooth}$ is the *i*-th point after smoothing, 2w is the window size, and $\frac{h_{i}}{H}$ is the smoothing coefficient, obtained by fitting polynomials by the least squares method.

### 4.3. Implementation Details

Our neural network is implemented based on Python 3.8 and Pytorch 1.10.1. We employ the last 80 images of the labeled 100-image dataset as the training set and the first 20 images as the test set. For all training data, random rotation and flipping are operated to diversify the data to improve the robustness of the model. CTS-Net is initialized using the CSWin Transformer pre-trained model on the ImageNet dataset. The hyperparameters $w_{1}$ and $w_{2}$ of the loss function are both set to 0.5, and *d* is set to 10. We set the epoch to 150 and the batch to 4 (because an original OCT image is divided into four images). The optimizer we use is Stochastic Gradient Descent (SGD) with an initial learning rate of 0.05, a momentum of 0.9, and a weight_decay of 1e-4. Furthermore, the change law of the learning rate is shown in Equation (Equation 7).
(7)$$lr=lr_{0}*{\left(1.0-\frac{it}{it_{\max}}\right)}^{0.95}$$
where lr is the current learning rate, $lr_{0}$ is the initial learning rate, it is the current iteration count, and $it_{max}$ is the maximum number of iterations.

### 4.4. Evaluation Metrics

The performance comparison analysis of the models uses three evaluation metrics, namely, mean absolute distance (MAD), root mean square error (RMSE), and dice-similarity coefficient (DSC), which are expressed by Equation (Equation 8). We compute the MAD and RMSE between the predicted and ground-truth edges along each column of pixels in the image.
(8)$$\begin{array}{c} \mathrm{MAD}=\frac{1}{N}\sum_{i}^{N-1}|p_{i}-g_{i}|\;\\ \mathrm{RMSE}=\sqrt{\frac{1}{N}\sum_{i}^{N-1}{\left(p_{i}-g_{i}\right)}^{2}}\;\\ \mathrm{DSC}=\frac{2\cdot |P\cap G|}{\left|P|+|G\right|}\times 100\% \end{array}$$
where *N* is the maximum number of columns; $p_{i}$ and $g_{i}$ denote the predicted and ground-truth edge positions, respectively; |P|, |G| represent the number of predicted and ground truth pixels; and |P∩G| represents the number of overlapping pixels between prediction and ground truth.

### 4.5. Experimental Results

#### 4.5.1. Analysis of Retinal Image Segmentation Results

To verify the effectiveness of our method, we compared it with three start-of-the-art models, namely, FCN [35], RelayNet [36], and Swin-Unet. FCN is a neural network model built with convolution as the backbone network and was employed to segment retinal layers. RelayNet is also a fully convolutional neural network model that can perform pixel-level segmentation of retinal layers. Swin-Unet is a U-shaped model with Swin Transformer as the backbone network, which first achieved good results in heart segmentation and multi-organ segmentation tasks and was later applied to various medical image segmentation tasks.

We compare the results of each model test for quantitative analysis, as shown in Table 1 and Table 2. Overall, the results of CTS-Net in the three evaluation metrics of MAD, RMSE, and DSC are better than those of other models. RelayNet ranks second in MAD and RMSE. CTS-Net is 0.31 and 0.48 pixels lower than RelayNet in terms of MAD and RMSE, respectively, and exceeds Swin-Unet (ranked 2) by 2.63% in DSC, which shows that it has outstanding performance.

A qualitative comparison between CTS-Net and other models found that it has smoother edges than other models and is closer to the ground truth, as shown in Figure 7. We observe that the edge CS, Swin-Unet, RelayNet, and FCN significantly differ from the ground truth labels, showing jagged, rough edges. As shown in Figure 8, CTS-Net also performs well in pixel-level segmentation. It can segment a narrow layer such as GCIPL well and ensure smooth edges of the area. For all six segmentation regions, CTS-Net significantly outperforms all compared methods.

From both quantitative and qualitative analyses, CTS-Net exhibits excellent performance, which can guarantee the topological structure of retinal layers and obtain smooth retinal boundaries. It has much to do with the model’s system design, the CSWin Transformer’s cross-attention mechanism, and the proposed loss function for edge regions.

#### 4.5.2. Loss Function Experimental Results

The boundary region loss function proposed in this paper has a hyperparameter *d*: the width of the edge region. We conduct several experiments to analyze this parameter’s influence on the model performance. On the premise of only adjusting the hyperparameter *d*, we train CTS-Net to obtain multiple models and test them to obtain the results in Table 3. In these five groups of experiments, the test results of MAD, RMSE, and DSC metrics are the best when *d* = 10. Compared with the five groups of experiments, the most significant differences in MAD, RMSE, and DSC metrics were 0.19 pixels, 0.22 pixels, and 0.13%, respectively. This experiment shows that the test performance of the model can be improved by selecting the hyperparameter *d* of the loss function.

To analyze the impact of the proposed loss function on the model’s performance, we select RelayNet, Swin-Unet, and CTS-Net to perform comparative experiments with different loss functions. There are four combinations of loss functions used in each model, namely, CrossEntropyLoss (CEL), CrossEntropyLoss + BoundaryAreaDiceLoss (CEL+BADL), DiceLoss (DL), and DiceLoss + BoundaryAreaDiceLoss (DL+BADL). The experiment results are shown in Table 4. After each model uses the loss function BADL, multiple evaluation metrics improved. The best combination of the loss function used in the RelayNet model is DL+BADL, which is 0.12 and 0.16 pixels lower in MAD and RMSE than the loss function used in the original paper, and 1.42% higher in DSC. Swin-Unet also improved in various evaluation metrics. The combination of CTS-Net’s loss function that performs best on MAD and RMSE is CEL+BADL, while the combination that performs best on the DSC is DL+BADL, but both cases are only slightly different. The results show that all the BADL loss functions will improve when the model is trained because BADL can realize further supervision on the edge area.

#### 4.5.3. Cross Validation

A stable and reliable model is essential; so, we use the cross-validation method to evaluate the robustness of CTS-Net. We perform a five-fold cross-validation experiment. Based on the dataset divided in Section 4.3, we split the 80 samples of the training set into five parts, one of which is the validation set, and the test set remains unchanged. Five models are obtained by training CTS-Net without repeated selection of the validation set, and the final results are shown in Table 5 and Table 6.

Overall, the test results of CTS-Net are similar for each fold because the ranges of MAD, RMSE, and DSC metrics are 0.05 pixels, 0.03 pixels, and 0.33%, respectively, with a slight difference. The test results of CTS-Net have slight differences because of different dataset partitions, which shows that the model has satisfactory robustness. The average values of MAD, RMSE, and DSC metrics of cross-validation results are better than those of FCN, RelayNet, and Swin-Unet, suggesting that CTS-Net has better generalization ability.

## 5. Discussion

Retinal layer segmentation for glaucoma is challenging due to the high resolution of the images involved and the high requirement for retinal layer edge segmentation. We designed CTS-Net based on the analysis of the cross-attention mechanism of the CSWin Transformer and the hierarchical characteristics of OCT retinal layer images. The proposed method is superior to the comparison model in MAD and RMSE metrics and inferior to the comparison model in Std, which shows that the test results of the model are relatively stable. Although the results of the BADL+DL loss function are similar to those of the Dice loss function (Table 4), when BADL is added, the MAD and RMSE metrics of the test results of the RelayNet, Swin-Net, and CTS-Net models are reduced, which indicates that BADL can guide the model to improve the performance of edge segmentation. It also shows that BADL can be used as a general loss function for image segmentation.

Our work also has some limitations. The proposed method must pre-process the rectangular image into a square before it can be used. However, Refs. [19,35,36] can directly input the rectangular image into the neural network. This problem can be solved by fine-tuning the CSWin Transformer structure to make it available for rectangular images. Our method is limited to 2D images and cannot process 3D data with rich information, which may be our future research work.

## 6. Conclusions

The neural network model based on CSWin Transformer has excellent potential in image segmentation but it is rarely used in OCT retinal layer image segmentation. The cross attention of CSWin Transformer can achieve a wide range of self-attention calculations, giving the model excellent remote modeling capability. In this paper, we built CTS-Net based on the basic skeleton of CSWin Transformer to realize pixel-level segmentation of OCT images of the glaucoma retinal layer and to obtain smooth and continuous retinal layer boundaries. We proposed a Dice loss function based on the boundary area to guide the model to discover more features around the edge region. The experimental results show that the evaluation indexes of MAD, RMSE, and DSC are 1.79 pixels, 2.15 pixels, and 92.79%, respectively, which are better than the contrast model. Further cross-validation experiments show the robustness of the CTS-Net method. The proposed method can effectively promote depth learning technology’s performance in retinal layer OCT image segmentation.

## Figures and Tables

**Figure 1 bioengineering-10-00230-f001:**
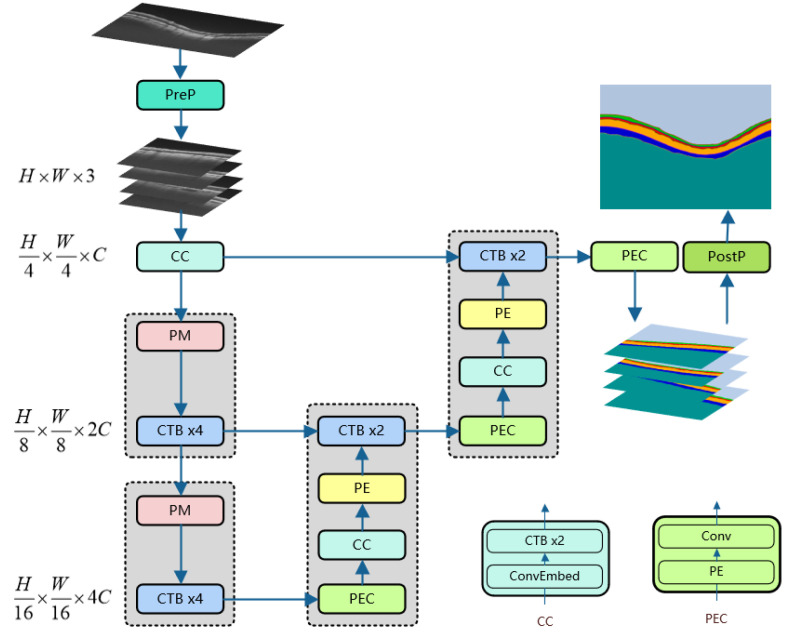
The architecture of CTS-Net. Among them, PE, CTB, PM, PreP, and PostP are the abbreviations of Patch Expanding, CSWin Transformer Block, Patch Merging, Pre-processing, and Post-processing, respectively.

**Figure 2 bioengineering-10-00230-f002:**
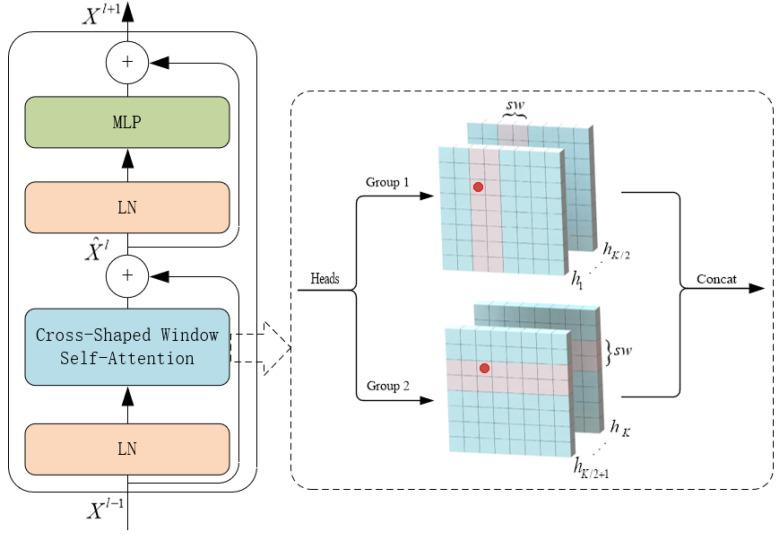
Structure of CSWin Transformer Block.

**Figure 3 bioengineering-10-00230-f003:**
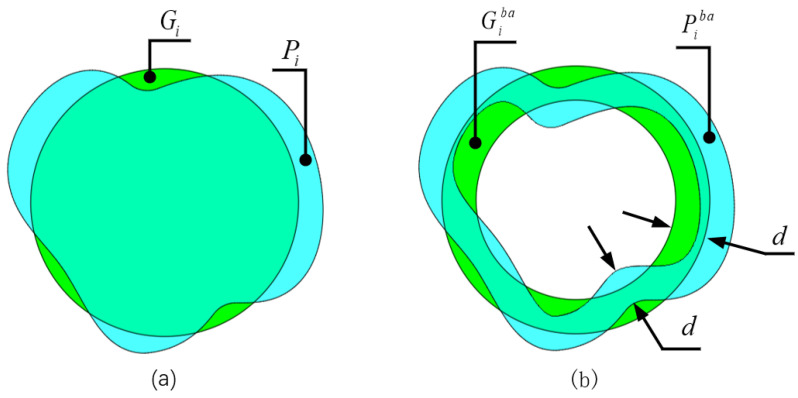
(**a**) The prediction results and ground truth of the *i*-th category. (**b**) The prediction result of the *i*-th class and the edge area of the ground truth; *d* is the width of the edge area.

**Figure 4 bioengineering-10-00230-f004:**
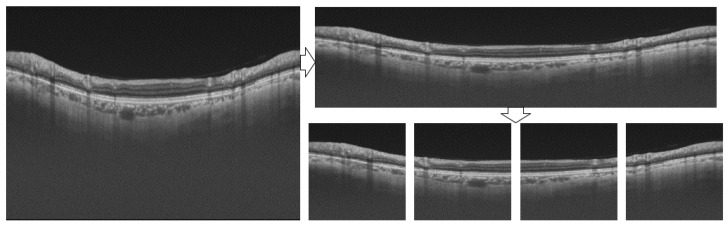
Image pre-processing flow.

**Figure 5 bioengineering-10-00230-f005:**
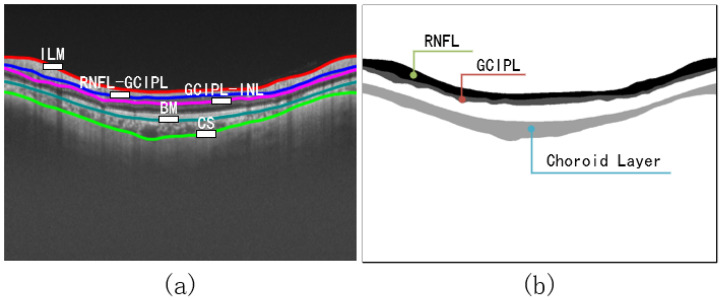
(**a**) OCT image with boundary naming. (**b**) Labels with instructions for retinal layers. The edge of each retinal layer in (**a**) is extracted from (**b**) using the Canny [33] algorithm.

**Figure 6 bioengineering-10-00230-f006:**
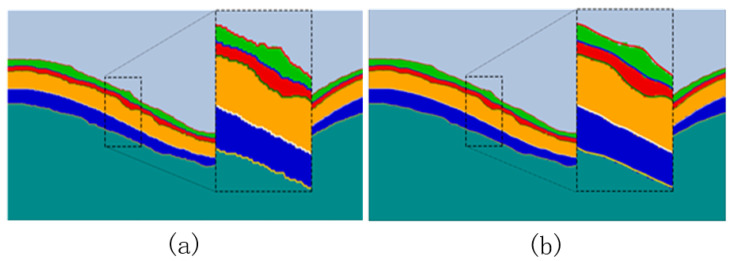
(**a**) Pixel-level prediction map extracted by Canny algorithm; (**b**) effect of the Savitzky–Golay filter algorithm to fit the edge.

**Figure 7 bioengineering-10-00230-f007:**
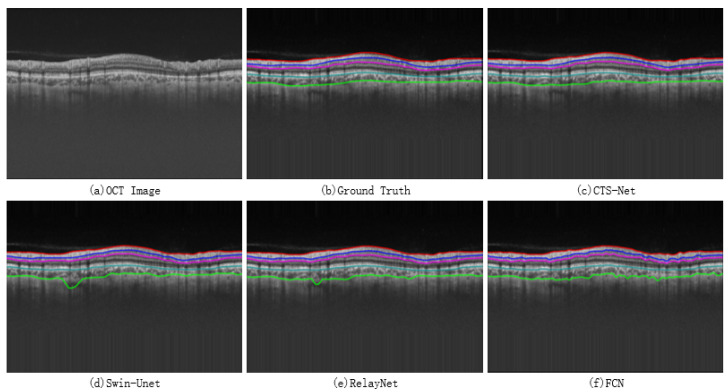
Retinal layer boundary results obtained by four methods.

**Figure 8 bioengineering-10-00230-f008:**
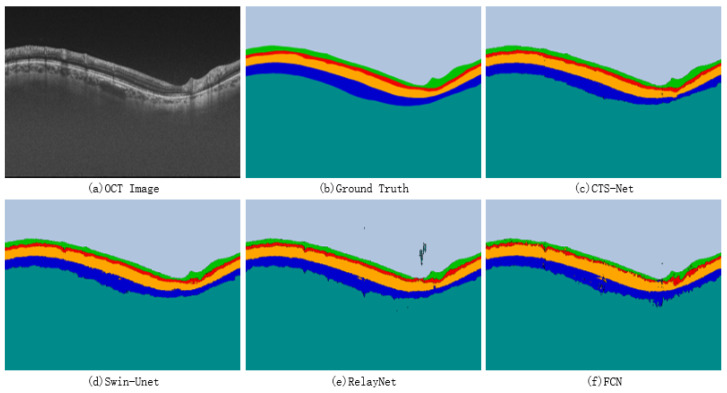
Retinal layer segmentation results for four methods.

**Table 1 bioengineering-10-00230-t001:** Test results of MAD(Std) and RMSE(Std) of each method in the GOALS dataset, where Std means standard deviation.

Boundary	MAD(Std)				RMSE(Std)			
	FCN	RelayNet	Swin-Unet	CTS-Net	FCN	RelayNet	Swin-Unet	CTS-Net
ILM	1.44(0.45)	**1.23(0.35)**	1.44(0.36)	1.40(0.41)	1.85(1.07)	**1.62(0.84)**	1.70(0.38)	1.63(0.41)
RNFL-GCIPL	2.24(0.69)	2.05(0.64)	1.97(0.59)	**1.47(0.36)**	3.05(0.98)	2.70(0.83)	2.53(0.76)	**1.86(0.45)**
GCIPL-INL	2.64(1.71)	1.84(0.72)	2.05(0.63)	**1.58(0.53)**	3.40(2.13)	2.29(0.83)	2.51(0.76)	**1.89(0.57)**
BM	1.76(0.52)	1.89(0.70)	1.97(0.60)	**1.73(0.65)**	2.38(0.95)	2.29(0.74)	2.41(0.68)	**2.06(0.71)**
CS	5.57(3.01)	3.51(1.27)	3.82(1.17)	**2.77(0.92)**	7.76(5.2)	4.25(1.46)	4.93(1.95)	**3.30(0.94)**
Overall	2.73(1.28)	2.10(0.74)	2.25(0.67)	**1.79(0.57)**	3.69(2.07)	2.63(0.94)	2.82(0.91)	**2.15(0.62)**

**Table 2 bioengineering-10-00230-t002:** Test results of DSC of each method in the GOALS dataset.

Layer	FCN	RelayNet	Swin-Unet	CTS-Net
RNFL	91.93%	93.06%	93.14%	**94.62%**
GCIPL	78.96%	84.88%	85.36%	**89.60%**
CL	89.98%	92.44%	91.98%	**94.14%**
Overall	86.96%	90.12%	90.16%	**92.79%**

**Table 3 bioengineering-10-00230-t003:** Quantitative comparison results of modifying the hyperparameter *d* of the boundary area loss function.

d	MAD(Std)	RMSE(Std)	DSC			
			RNFL	GCIPL	CL	Overall
6	1.91(0.65)	2.28(0.71)	94.60%	88.81%	93.79%	92.40%
8	1.97(0.65)	2.35(0.69)	**94.68%**	88.71%	93.57%	92.32%
10	**1.83(0.61)**	**2.22(0.68)**	94.50%	**89.01%**	**93.81%**	**92.44%**
12	2.02(0.70)	2.44(0.79)	94.63%	88.90%	93.39%	92.31%
14	1.98(0.63)	2.37(0.68)	94.56%	88.86%	93.61%	92.34%

**Table 4 bioengineering-10-00230-t004:** The results of the model using different combinations of loss functions.

Model	Loss Function	MAD(Std)	RMSE(Std)	DSC			
				RNFL	GCIPL	CL	Overall
RelayNet	CEL	2.10(0.69)	2.58(0.79)	93.09%	84.51%	92.44%	90.01%
	CEL+BADL	2.04(0.66)	2.69(1.31)	93.39%	86.37%	92.54%	90.77%
	DL	1.98(0.74)	2.56(1.40)	**93.85%**	87.26%	**93.08%**	91.40%
	DL+BADL	**1.98(0.66)**	**2.47(1.06)**	94.08%	**87.81%**	92.83%	**91.58%**
Swin-Unet	CEL	2.64(0.75)	3.44(1.21)	90.92%	79.35%	88.69%	86.32%
	CEL+BADL	2.16(0.64)	2.71(0.85)	93.19%	85.25%	91.84%	90.09%
	DL	2.17(0.64)	2.68(0.80)	93.56%	86.21%	**92.39%**	90.72%
	DL+BADL	**2.13(0.61)**	**2.65(0.88)**	**93.65%**	**86.73%**	92.34%	**90.91%**
CTS-Net	CEL	2.03(0.64)	2.45(0.69)	94.12%	87.48%	93.40%	91.67%
	CEL+BADL	**1.78(0.55)**	**2.14(0.60)**	94.62%	89.35%	93.80%	92.59%
	DL	1.82(0.58)	2.19(0.61)	**94.80%**	89.43%	93.97%	92.73%
	DL+BADL	1.79(0.57)	2.15(0.62)	94.62%	**89.60%**	**94.14%**	**92.79%**

**Table 5 bioengineering-10-00230-t005:** Test results of MAD and RMSE with five-fold cross-validation.

Boundary	MAD(Std)					
	K = 1	K = 2	K = 3	K = 4	K = 5	Mean
ILM	1.30(0.36)	1.23(0.36)	1.41(0.40)	1.31(0.37)	1.36(0.38)	1.32(0.37)
RNFL-GCIPL	1.67(0.39)	1.71(0.48)	1.69(0.44)	1.79(0.48)	1.73(0.45)	1.72(0.45)
GCIPL-INL	1.73(0.54)	1.90(0.61)	1.53(0.53)	1.75(0.58)	1.58(0.53)	1.70(0.56)
BM	1.65(0.49)	1.67(0.52)	1.69(0.53)	1.67(0.54)	1.74(0.64)	1.68(0.54)
CS	3.12(1.07)	3.05(0.99)	3.03(1.14)	2.96(1.04)	2.90(0.86)	3.01(1.02)
Overall	1.89(0.57)	1.91(0.59)	1.87(0.61)	1.90(0.60)	1.86(0.57)	1.89(0.59)
	**RMSE(Std)**					
	**K = 1**	**K = 2**	**K = 3**	**K = 4**	**K = 5**	**Mean**
ILM	1.52(0.37)	1.47(0.38)	1.64(0.39)	1.53(0.37)	1.60(0.38)	1.55(0.38)
RNFL-GCIPL	2.08(0.53)	2.12(0.64)	2.09(0.58)	2.19(0.61)	2.15(0.61)	2.13(0.59)
GCIPL-INL	2.05(0.57)	2.23(0.64)	1.86(0.59)	2.08(0.61)	1.92(0.55)	2.03(0.59)
BM	1.99(0.55)	2.01(0.59)	2.04(0.57)	2.00(0.60)	2.06(0.69)	2.02(0.60)
CS	3.75(1.20)	3.59(0.97)	3.61(1.18)	3.56(1.11)	3.55(0.99)	3.61(1.09)
Overall	2.28(0.64)	2.28(0.64)	2.25(0.66)	2.27(0.66)	2.26(0.64)	2.27(0.65)

**Table 6 bioengineering-10-00230-t006:** Test results of DSC with five-fold cross-validation.

Layer	K = 1	K = 2	K = 3	K = 4	K = 5	Mean
RNFL	94.72%	94.75%	94.68%	94.71%	94.55%	94.68%
GCIPL	89.42%	89.12%	89.22%	89.04%	88.65%	89.09%
CL	93.85%	93.68%	93.78%	93.87%	93.78%	93.79%
Overall	92.66%	92.52%	92.56%	92.54%	92.33%	92.52%

## Data Availability

The public dataset is available in a publicly accessible repository. Please see [30].

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
