# Peer review of "CTS-Net: A Segmentation Network for Glaucoma Optical Coherence Tomography Retinal Layer Images"

_bioengineering, 2023, doi:10.3390/bioengineering10020230_

Round 1
Reviewer 1 Report
It is appreciated that the authors proposed a new method based on CSWin Transformer to segment OCT images with high accuracy. However, my major concern is that only 100 images were used and no external testing to show whether the algorithm is overfitted or can also generalize to other unseen dataset. Besides, although the images were from a competition announced online, the authors should provide more details of the ground truth, i.e., whether the images were macula OCT or circumpapillary OCT, how many are labelled as glaucoma, how many involved the optic nerve head area, etc. Specific comments are as follows.
1. Abstract: “It is very time-consuming for ophthalmologists to directly analyze OCT images because the data collected by OCT equipment is enormous, so there is an urgent need for automatic segmentation technology.” The need of automatic segmentation technology is not due to difficulty in OCT image analysis, but due to the potential segmentation error using OCT built-in software. Besides, the more important thing is to accurately quantify the segmented layers and provide metrics to evaluate the structural changes.
2. Same comment on the Introduction as on the Abstract. The authors didn’t describe the importance of OCT segmentation and the limitation of existing methods.
3. Line 86: “The underlying retinal images are generally grayscale images collected through X-ray, CT, MRI, and OCT technologies.” What do the authors mean? Retinal images are definitely not collected through X-ray, CT, MRI.
4. Line 232: The authors should give more information on the OCT dataset and the ground truth labelling as I mentioned before.
5. Line 270 and Line 342: Only 100 images were used for the training, without comparison with a model trained with large dataset or testing on unseen dataset. It is not scientific enough to claim that it can achieve outstanding performance on small datasets. It could be overfitted or maybe a model with large sample size can perform even better.
6. The discussion part should have more in-depth discussion of comparison with other studies/methods, the strengths and limitations of the proposed method.
Reviewer 2 Report
The authors are suggested the following changes
1. Objective of the research work needs to be added.
2. Comparative-related work required.
3. Proposed work elaboration required.
3. What is hyper-parameter tunning with loss function? Justify.
4. Authors need to justify results with more evaluation parameters.
5. Elaborate more SGD optimizer in the implementation part.
6. Use cross-fold validation techniques in the experimentation part.
7. Authors need to rewrite the conclusion part for clarity about the proposed article.
Reviewer 3 Report
I read the paper entitled ”A Segmentation Network for Glaucoma Optical Coherence Tomography Retinal Layer Images” very carefully and concluded that the paper is acceptable in the present form for publication in your journal. The topic of the article is interesting. The paper is very good structured and exact including figures and tables and contributes contributes to further understanding for glaucoma patients.
Round 2
Reviewer 1 Report
The authors addressed my previous comments. I have no further comments.
Reviewer 2 Report
THE MANUSCRIPT WAS MODIFIED AS PER THE REQUIREMENT OF THE COMMENTS